# Light-dependent pathways for dopaminergic amacrine cell development and function

**Teona Munteanu[1†], Katelyn J Noronha[1], Amanda C Leung[1], Simon Pan[2], Jasmine A Lucas[1], Tiffany M Schmidt[1]***

[1]Department of Neurobiology, Northwestern University, Evanston, United States; [2]Department of Biology, Johns Hopkins University, Baltimore, United States

**Abstract** Retinal dopamine is a critical modulator of high acuity, light-adapted vision and photoreceptor coupling in the retina. Dopaminergic amacrine cells (DACs) serve as the sole source of retinal dopamine, and dopamine release in the retina follows a circadian rhythm and is modulated by light exposure. However, the retinal circuits through which light influences the development and function of DACs are still unknown. Intrinsically photosensitive retinal ganglion cells (ipRGCs) have emerged as a prime target for influencing retinal dopamine levels because they costratify with DACs in the inner plexiform layer and signal to them in a retrograde manner. Surprisingly, using genetic mouse models lacking specific phototransduction pathways, we find that while light influences the total number of DACs and retinal dopamine levels, this effect does not require ipRGCs. Instead, we find that the rod pathway is a critical modulator of both DAC number and retinal dopamine levels.
DOI: https://doi.org/10.7554/eLife.39866.001

*For correspondence:
tiffany.schmidt@northwestern.edu

†These authors contributed equally to this work

**Competing interests:** The authors declare that no competing interests exist.

## Introduction

Dopamine is released in the retina by dopaminergic amacrine cells (DACs), and follows a circadian rhythm with high dopamine levels occurring during the day and low levels at night (*Doyle et al., 2002*). Retinal dopamine plays a role in resetting the retinal circadian clock and serves as a regulator of high acuity, light-adapted vision (*Besharse and McMahon, 2016*; *Jackson et al., 2012*; *Prigge et al., 2016*) and photoreceptor coupling in the retina (*Jin et al., 2015*; *Ribelayga et al., 2008*). For example, mice lacking tyrosine hydroxylase (TH), the rate limiting enzyme in dopamine synthesis, show deficits in contrast sensitivity, acuity, and retinal circadian rhythms that are reversed by administration of L-DOPA (*Jackson et al., 2012*). DACs receive light input from rods, cones, and the melanopsin-expressing intrinsically photosensitive retinal ganglion cells (ipRGCs). DACs have been shown to respond to light through the rod pathway, and the rod pathway has been implicated in DA regulation. Likewise, DACs respond to light through the cone pathway (*Zhang et al., 2008*; *Zhao et al., 2017*). ipRGC dendrites co-stratify with DACs within the IPL during development and adulthood (*Matsuoka et al., 2011*; *Vugler et al., 2007*), and they signal in a retrograde manner to DACs (*Zhang et al., 2008*; *Zhang et al., 2012*), which has made them a likely candidate for regulation of retinal dopamine.

Despite the importance of light signaling in regulating dopamine levels, the cell types and circuits through which light influences DAC development and function have not been identified. We therefore sought to identify whether and through which photoreceptive pathways light exerts these effects. We find that light exposure is indeed required to set the number of TH-positive (TH+) cells and dopamine levels in the retina, and that rod phototransduction is critical for this process. Surprisingly, though ipRGCs have arisen as a prime candidate for regulation of dopamine in the retina,

retinas lacking melanopsin phototransduction or with ipRGCs ablated show normal dopamine levels and TH+ cell number. Collectively, our results indicate that rod signaling, through cell types other than ipRGCs, is primarily responsible for the effects of light on DAC development.

## Results

### Light exposure during development sets TH-positive cell number and dopamine levels

We first tested whether exposure to a 12:12 light:dark cycle influences the number of TH+ amacrine cells in the retina. To do this, we performed immunohistochemistry for TH and compared the number of TH+ cells in the retinas of animals that were reared in constant darkness from conception to those reared on standard 12:12 cycles (*Figure 1D*). Rearing in constant darkness resulted in decreased numbers of TH +cells at P14 (DD: Mean ± SEM = 580.8 ± 8.6 cells, n = 15 retinas; LD: Mean ± SEM = 643.7 ± 8.2 cells, n = 6 retinas; Unpaired t-test, p = 0.0004) (*Figure 1A*). This decrease persisted when animals were dark-reared to adult stages (DD: Mean ± SEM = 567.3 ± 10.2 cells, n = 6 retinas; LD: Mean ± SEM = 648.5 ± 17.5 cells, n = 6 retinas; One-Way ANOVA, p = 0.0002, Dunnett's multiple comparisons test) (*Figure 1B*). Interestingly, we found that this

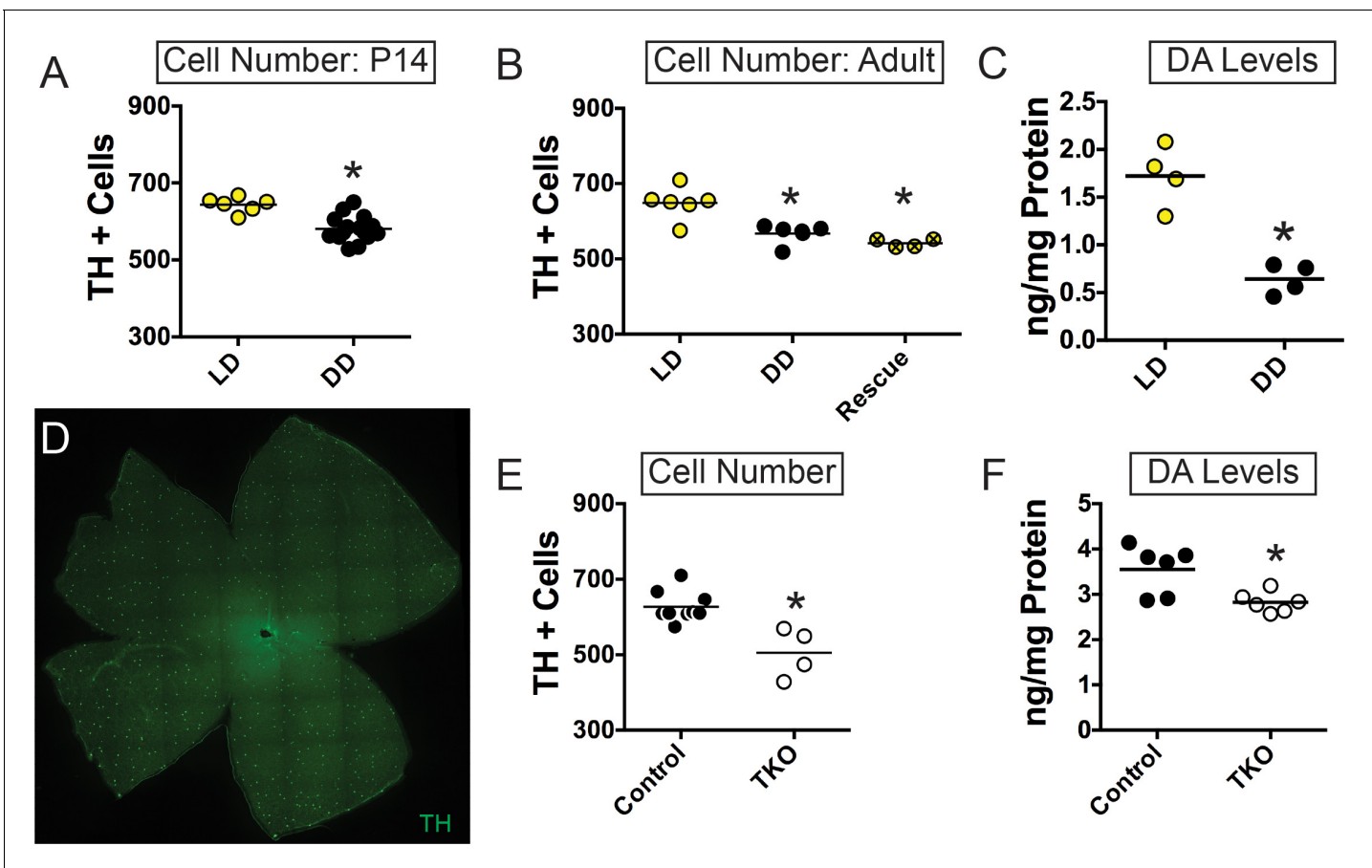

**Figure 1.** Light exposure during early retinal development is required to set TH-positive cell number and dopamine levels. (**A–B**) TH+ cell number in animals bred and reared in either LD (yellow circles) or DD (black circles) from conception at P14 (n = 6 LD, n = 15 DD) (**A**) or Adult (n = 6 LD, n = 6 DD) (**B**) stages. Adult TH+ cell number could not be rescued by moving animals from DD to LD at P14 (hatched yellow circles, n = 4) (**C**) DA levels in adult retinas from animals reared in LD (yellow circles, n = 4) or DD (black circles, n = 4). (**D**) TH+ cells in whole mount WT adult retina (**E–F**) TH+ cell number (n = 9 Control, n = 4 TKO) (**E**) and DA levels (n = 6 Control, n = 6 TKO) (**F**) in Control (black circles) and TKO (open circles) adult retinas. *p < 0.05. Bars on plots represent mean.

DOI: https://doi.org/10.7554/eLife.39866.002

decrease in TH+ cell number in the adult retina persisted even after dark-reared mice were placed in LD at P14, demonstrating that this decrease in TH+ cell number could not be rescued by subsequent exposure to a normal LD cycle after eye opening (LD Rescue: Mean ± SEM = 541.8 ± 5.6 cells, n = 4 retinas; One-Way ANOVA, Dunnett's multiple comparisons test) (*Figure 1B*). In support of decreased TH+ cell number, dopamine (DA) levels were significantly reduced in DD reared animals (DD: Mean ± SEM = 0.6 ± 0.08 ng/mg, n = 4 retinas; LD: Mean ± SEM = 1.723 ± 0.16 ng/mg, n = 4 retinas; Unpaired t-test, p = 0.003) (*Figure 1C*).

If light signaling is occurring through one of the three known photoreceptors (rods, cones, or ipRGCs), then rendering all three of these photoreceptor types insensitive to light through genetic perturbation of phototransduction cascade components (*Gnat1$^{-/-}$; Gnat2$^{cpfl3/cpfl3}$; Opn4$^{-/-}$*, hereafter referred to as TKO for 'triple knockout') should mimic the phenotype we observed in DD reared animals. Indeed, TKO retinas had significantly reduced number of TH+ cells (TKO: Mean ± SEM = 505.0 ± 32.8 cells, n = 4 retinas; Control: Mean ± SEM = 627.4 ± 13.5 cells, n = 9 retinas; One-Way ANOVA, p < 0.0001, Dunnett's multiple comparisons test) and significantly reduced DA levels (TKO: Mean ± SEM = 2.82 ± 0.09 ng/mg, n = 6 retinas; Control: Mean ± SEM = 3.55 ± 0.22 ng/mg, n = 6 retinas; One-Way ANOVA, p < 0.0001, Dunnett's multiple comparisons test) compared to control retinas (*Figure 1E–F*). These results indicate that light is a critical modulator of TH+ cell number and resting DA levels in the retina during development and into adulthood.

## Rod, but not cone or melanopsin, phototransduction influence TH-positive cell number and dopamine levels

We next wanted to determine which photoreceptor types are involved in setting TH+ cell number and DA levels during development (*Figure 2A*). To do this, we counted TH+ cells in retinas from littermate mice lacking functional rod (*Gnat1$^{-/-}$*, hereafter referred to as RKO), cone (*Gnat2$^{cpfl3/cpfl3}$*, hereafter referred to as CKO), or melanopsin (*Opn4$^{-/-}$*, hereafter referred to as MKO) phototransduction cascades (*Figure 2B*). Despite reports of ipRGC influence on DA-dependent retinal physiology (*Prigge et al., 2016*; *Zhang et al., 2008*; *Zhao et al., 2017*; *Zhang et al., 2012*; *Qiao et al., 2017*), we found that MKO and CKO mice showed no significant change in TH+ cell number (MKO: Mean ± SEM = 615.3±55.5 cells, n = 6 retinas; CKO: Mean ± SEM = 708.6 ± 30.74 cells, n = 9 retinas; Control Mean ± SEM = 627.4 ± 13.5 cells, n = 9 retinas; One-Way ANOVA, Dunnett's multiple comparisons test) or DA levels (MKO: Mean ± SEM = 3.63 ± 0.15 ng/mg, n = 6 retinas; CKO: Mean ± SEM = 3.88 ± 0.07 ng/mg, n = 6 retinas; Control Mean ± SEM = 3.55 ± 0.22 ng/mg, n = 6 retinas; One-Way ANOVA, Dunnett's multiple comparisons test), though CKO mice showed a trend toward increasing numbers of TH+ cells (*Figure 2C–D*). These data indicate that lack of melanopsin or cone phototransduction does not influence basal retinal DA as previously suggested (*Dkhissi-Benyahya et al., 2013*). Surprisingly, RKO mice showed a significant decrease in both TH+ cell number (RKO: Mean ± SEM = 518.5 ± 29.3 cells, n = 9; Control Mean ± SEM = 627.4 ± 13.5 cells, n = 9; One-Way ANOVA, p < 0.0001, Dunnett's multiple comparisons test) and DA levels (RKO: Mean ± SEM = 2.92 ± 0.09 ng/mg, n = 6 retinas; Control Mean ± SEM = 3.55 ± 0.22 ng/mg, n = 6 retinas; One-Way ANOVA, p < 0.0001, Dunnett's multiple comparisons test). Our data from RKO tracks closely with our results from DD reared animals and TKO animals (*Figure 1*), indicating that rods in particular play a critical role in setting TH+ cell number and DA levels in the developing and adult retina.

## Rod influence on retinal dopamine and TH-positive cell number does not occur through ipRGCs

Rods and cones signal to DACs through bipolar cells (*Zhang et al., 2008*; *Qiao et al., 2016*) as well as in a retrograde fashion via ipRGCs (*Zhang et al., 2008*; *Zhao et al., 2017*; *Zhang et al., 2012*). Thus, it is possible that though melanopsin phototransduction itself is not required for normal TH + cell number and retinal DA levels, that the rod-based signals are still reaching DACs via ipRGCs (*Figure 2A*). If this is the case, then ablating ipRGCs during development should phenocopy our results from RKO animals (*Figure 2B*). To test this, we counted TH+ cells and measured DA levels in retinas from animals where ipRGCs are ablated early in development (*Opn4$^{DTA/DTA}$*, hereafter referred to as DTA animals) (*Figure 2B*) (*Chew et al., 2017*). Surprisingly, ablation of ipRGCs did

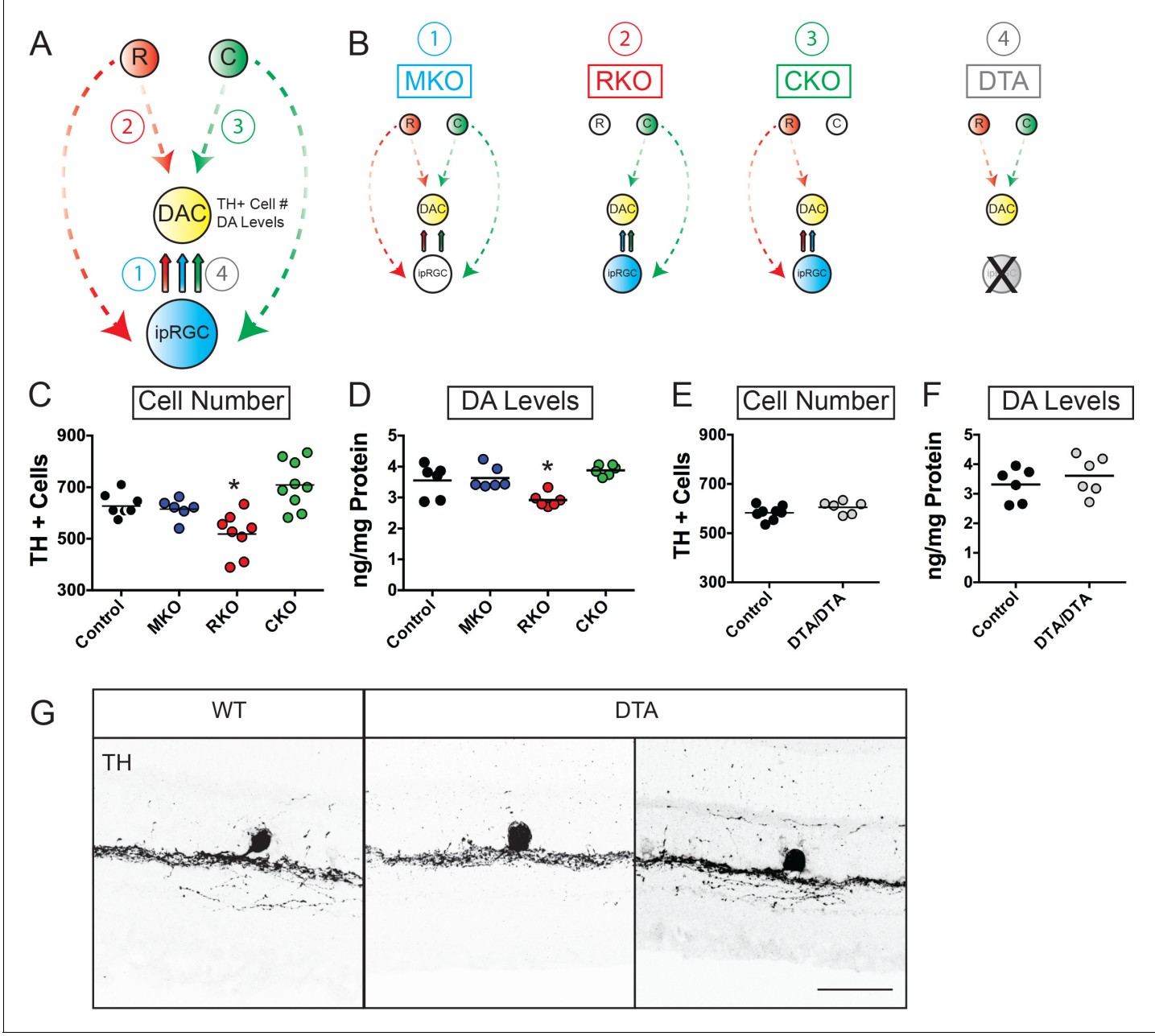

**Figure 2.** Rod signaling influences TH-positive cell number and dopamine levels through ipRGC-independent pathways. (**A**) Diagram depicting potential pathways by which light information could reach DACs (1. Via melanopsin signals relayed by ipRGCs, (2. Via rod signals relayed to DACs, (3. Via cone signals relayed to DACs, or (4. Via rod, cone, and/or melanopsin signals through ipRGC-dependent pathways. Dashed arrows represent indirect influence through multicellular circuits while solid arrows represent direct synaptic connectivity between subtypes. (**B**) Diagram depicting signaling pathways disrupted in MKO, RKO, CKO, and DTA mouse lines. Dashed arrows represent indirect influence through multicellular circuits while solid arrows represent direct synaptic connectivity between subtypes. (**C–D**) TH+ cell number and DA levels in Control (black circles, n = 9), MKO (blue circles, n = 6), RKO (red circles, n = 8), and CKO (green circles, n = 9) retinas from adult littermates. (**E–F**) TH+ cell number (n = 8 Control, n = 6 DTA) and DA levels (n = 6 Control, n = 6 DTA) in Control (black circles) and DTA (gray circles) retinas from adult littermates. (**G**) TH+ cell anatomy in WT and DTA adult retinal sections. We observed no morphological differences between TH+ cells in these two mouse lines. DA: dopamine, DAC: dopaminergic amacrine cell, MKO: animals lacking melanopsin phototransduction, RKO: animals lacking rod signaling, CKO: animals lacking cone signaling, DTA: animals where ipRGCs are ablated through expression of diphtheria toxin. Scale bar in (**G**) is 50 µm. *p < 0.05, bars on plots represent mean.

DOI: https://doi.org/10.7554/eLife.39866.003

The following figure supplement is available for figure 2:

*Figure 2 continued*

**Figure supplement 1.** Examples of TH-positive cell stratification in WT and DTA retinas.

DOI: https://doi.org/10.7554/eLife.39866.004

not affect the number of TH+ cells (DTA: Mean ± SEM = 605.2 ± 10.5 cells, n = 6 retinas; Control: Mean ± SEM = 582.6 ± 10.1 cells, n = 8 retinas; Unpaired t-test, p = 0.1522) or retinal DA levels (DTA: Mean ± SEM = 3.61 ± 0.27 ng/mg, n = 6 retinas; Control: Mean ± SEM = 3.31 ± 0.23 ng/mg, n = 6 retinas; Unpaired t-test, p = 0.4188), indicating that ipRGCs are not the relay for rod-dependent signals influencing these processes (*Figure 2E–F*). Additionally, we find that TH+ cell stratification also appears normal in DTA retinas, providing further support for normal DAC development in the absence of ipRGCs (*Figure 2G* and *Figure 2—figure supplement 1*). These data indicate that rod phototransduction through non-ipRGC pathways is the critical regulator of both TH+ cell number and resting retinal DA levels (*Figure 3*).

## Discussion

Our findings indicate that rods are the major photoreceptor class in the retina regulating the number of TH+ cells as well as resting retinal DA levels, and that they are doing so through ipRGC-independent pathways (*Figure 3*). These findings are consistent with reports that DACs receive rod/cone pathway input relayed via bipolar cells (*Zhao et al., 2017*; *Qiao et al., 2016*). Specifically, a recent report demonstrated that rod responses were consistently detectable in electrophysiological recordings from DACs across multiple luminance levels. Cone responses, however, were detectable at lower light intensities in the dorsal versus ventral retina, while melanopsin-based responses were detectable only in the dorsal retina (*Zhao et al., 2017*). Our results in DD and RKO animals are roughly parallel, suggesting that rods are the primary drivers of the defects observed in DD. We did observe a proportionally larger DA decrease in DD, which could indicate that there are other pathways at play. However, this larger DA decrease could also be due to the higher light levels that the LD animals were reared in compared to the Controls in the RKO experiment. RKO animals have been shown to have some retinal degeneration starting at 4 months, which could also explain the differences in magnitude (*Calvert et al., 2000*). This is unlikely because dark-reared animals do not have retinal degeneration, but still exhibit reduced TH+ cell number and decreased retinal DA levels (*Brooks et al., 2014*). It is also possible that these differences were due to slight genetic variability amongst cohorts because the number of DACs is closely tied to genetic background, though both of these groups were maintained on the same mixed hybrid background (see Materials and methods) (*Whitney et al., 2009*). *Gnat2*$^{cpfl3/cpfl3}$ ('CKO') mice have been shown to retain some low sensitivity cone function (*Chang et al., 2006*), and our CKO mice showed a non-significant trend toward increased DAC number, indicating that cones may have some influence on DAC number. Nevertheless, because only RKO or TKO animals recapitulated the phenotype we observed in DD, our data argue that rods are the major pathway through which this occurs.

That ipRGCs were not required for either TH+ cell number or setting retinal DA levels was surprising given the close apposition of DAC axons and ipRGC dendrites, as well as evidence for bidirectional signaling between these two populations and effects of melanopsin signaling on the number of other retinal cell types (*Matsuoka et al., 2011*; *Vugler et al., 2007*; *Zhang et al., 2012*; *Rao et al., 2013*). Previous work has shown deficits in the light adapted ERG of DTA animals that can be rescued by a D4 agonist (*Prigge et al., 2016*). There are two possible explanations for this deficit: The first is that there are differences in evoked DA release in acute light exposure but not in resting DA. As our measurements of DA were taken from light adapted animals during the animal's subjective day, this seems unlikely. Our recent work points to a second possible explanation. We have found that during development, DTA retinas show a down regulation of the D4 receptor, Drd4, which is expressed on cone photoreceptors (*Tufford et al., 2018*). Therefore, it is possible that lack of ipRGCs causes genetic alterations in the retina of DTA animals that makes individual cell populations in the retina less responsive to DA. However, we have not examined Drd4 expression levels in adult DTA versus WT retinas.

It is important to note that while our data clearly indicate fewer TH+ cells in RKO and DD retinas, we were limited to the use of TH immunopositive cells as a proxy for DAC number because other

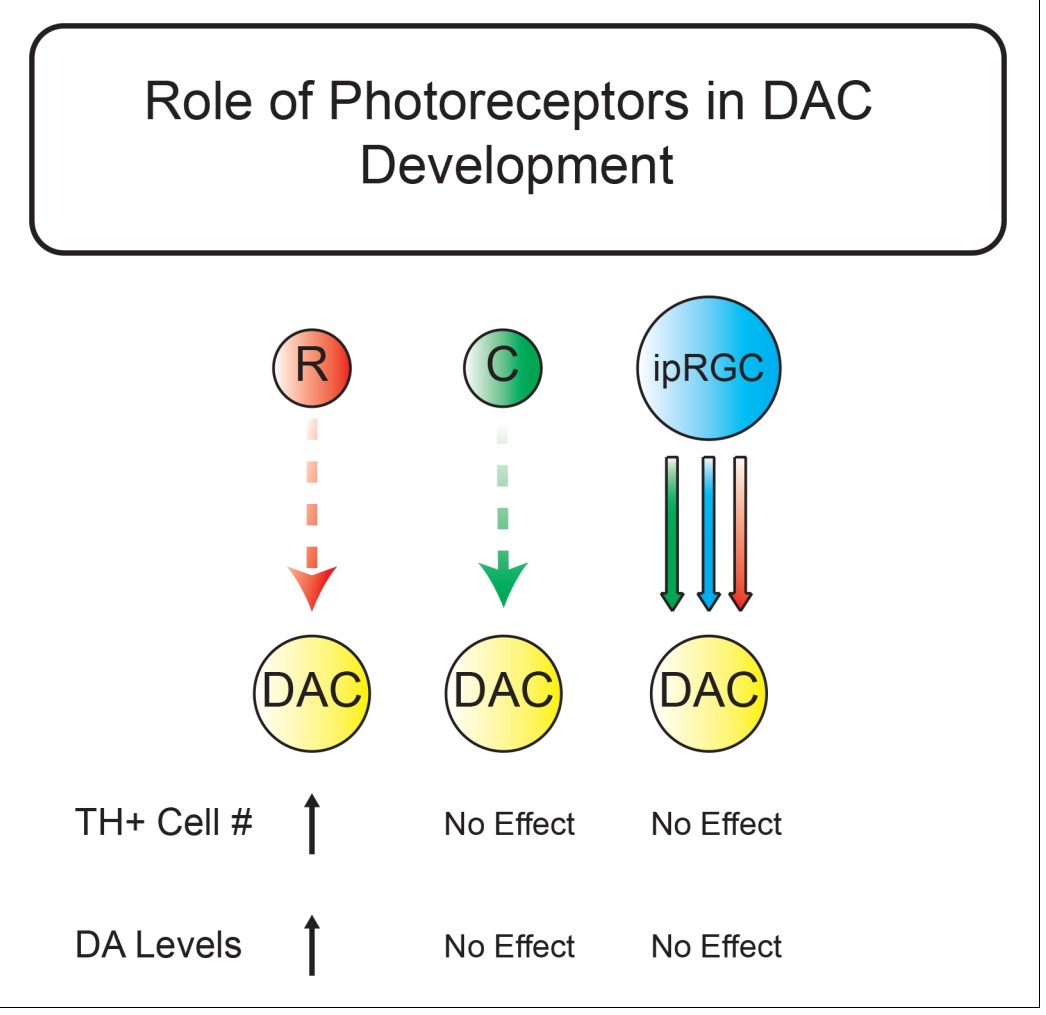

**Figure 3.** Summary schematic for light-dependent influences on TH-positive cell number and retinal dopamine levels. Rods signaling serves to increase TH+ cell number and DA levels through ipRGC-independent pathways. Dashed arrows represent indirect influence through multicellular circuits while solid arrows represent direct synaptic connectivity between subtypes. Red represents rod pathway signals, green represents cone pathway signals, and blue represents signals arising from melanopsin phototransduction. Neither ipRGC relay of rod, cone, nor melanopsin signals nor the cone pathway influence TH+ cell number or retinal DA levels. The rod pathway serves to increase the number of TH+ cells and increase retinal DA levels. R: Rod, C: Cone, DAC: dopaminergic amacrine cell, DA: Dopamine.
DOI: https://doi.org/10.7554/eLife.39866.005

currently available dopamine reporter reagents label additional and varied cell types in the retina (*Vuong et al., 2015*). Therefore, we were unable to determine whether the decrease in TH + cell number truly represents a decreased number of DACs via either cell death or lack of differentiation, or whether decreased cell numbers simply represent undetectable TH protein levels in a subset of DACs. However, in the absence of sufficient TH enzyme, any surviving DACs will be unable to functionally synthesize dopamine. Thus, though we cannot clarify the reasons for the absence of TH immunolabeling at this time, it is clear that dopamine production from a subset of DACs is lost or at least severely attenuated in DD and RKO mice. In the future it will be to determine whether the actual number of DACs in RKO animals is in fact lower than in WT, or whether the levels of TH are simply undetectable in some DACs.

# Materials and methods

## Key resources table

| Reagent type (species) or resource | Designation | Source or reference | Identifiers | Additional information |
|---|---|---|---|---|
| Genetic reagent (*M. musculus*) | *Opn4*<sup>LacZ/LacZ</sup> | PMID: 11834834 | RRID: MGI:3797748 | |
| Genetic reagent (*M. musculus*) | *Gnat2*<sup>cpfl3/cpfl3</sup> | PMID: 17065522 | RRID: MGI:3715214 | |
| Genetic reagent (*M. musculus*) | *Gnat1*<sup>-/-</sup> | PMID: 11095744 | RRID: MGI:3640094 | |
| Genetic reagent (*M. musculus*) | *Opn4*<sup>DTA/DTA</sup> | PMID: 28617242 | n/a | |
| Antibody | Rabbit anti-TH | Millipore | Cat: AB152 | IHC (1:500) |

## Animals

All procedures were approved by the Animal Care and Use Committee at Northwestern University (Protocol number IS00000887). Both male and female mice were used in this study. For LD and DD experiments, F1 C57Bl6/J; 129S1/SvlmJ wild-type mice from Jackson Labs were bred to generate hybrid progeny. For manipulations of specific phototransduction cascades, mouse lines were generated from the following cross: $Opn4^{+/-}$; $Gnat1^{+/-}$; $Gnat2^{+/cpfl3}$ X $Opn4^{+/-}$; $Gnat1^{+/-}$; $Gnat2^{+/cpfl3}$. Each locus outside the locus of interest was either heterozygous or wild-type. Control mice were either heterozygous or wild-type at a given locus. From this cross were able to generate all MKO ($Opn4^{-/-}$), RKO ('rod knock out,' $Gnat1^{-/-}$), and CKO ('cone knock out,' $Gnat2^{cpfl3/cpfl3}$) mice (*Calvert et al., 2000*; *Chang et al., 2006*; *Altimus et al., 2010*; *Hattar et al., 2002*) but only a single TKO ('triple knockout,' $Opn4^{-/-}$; $Gnat1^{-/-}$; $Gnat2^{cpfl3/cpfl3}$) mouse due to low probability of homozygosity at every locus. We therefore used offspring from this initial mating to generate all other TKO mice used in the study. To generate $Opn4^{DTA/DTA}$ and the associated controls, we crossed $Opn4^{DTA/+}$ X $Opn4^{DTA/+}$ and utilized $Opn4^{DTA/DTA}$ (DTA) (*Chew et al., 2017*) and $Opn4^{+/+}$ (Control) littermates generated from this cross for all experiments.

For LD/DD experiments animals were bred and reared in a 12:12 light:dark cycle (Light intensity ~2000 lux) except in the case of the dark-reared animals, which were bred and reared from conception in light-tight chambers. Some of the dark-reared animals were placed in the Rescue group. These animals were bred and maintained in light-tight chambers until P14 and then moved to a 12:12 LD cycle (Light intensity ~2000 lux). For DD conditions, cages were changed using very dim, red illumination. For mutant studies, animals were housed in the animal facility where the light intensity was 100 – 200 lux. All animals received food and water ad libitum.

## Immunohistochemistry

Mice were euthanized at postnatal day 14 (P14) and at P50-90. Whole retinas were isolated and placed in 4% paraformaldehyde (PFA) at 4°C. Following a 1 hr incubation, retinas were then washed with phosphate buffered saline (PBS) three times for 5 min each at room temperature (RT). Blocking solution consisted of 6% goat serum in 0.3% Triton (Millipore) in PBS. Retinas were kept in blocking solution for 1 hr at RT. Retinas were then placed in primary antibody solution overnight at 4°C, at a 1:500 dilution of rabbit anti TH antibody (Millipore) in 0.3% Triton. Retinas were once again washed in PBS, and then placed in a 1:500 Alexa fluor 488 goat anti-rabbit IgG (Invitrogen) secondary antibody solution for 2 to 4 hr at RT. Retinas were washed once again and mounted onto glass microscope slides, and the coverslip was secured using Fluoromount. Slides were then stored at 4°C until imaged.

For retinal sections, mice were euthanized at P14 or P30 and the eyes were removed. The cornea and lens were removed, and the isolated eyecups were fixed in 4% PFA at 4°C for 2 hr and then cryoprotected in 30% sucrose overnight. Eyecups were then mounted in OCT freezing media and stored at −20°C overnight. The retinal tissue was then sectioned at 16 μm on a Leica CM1950 Cryostat and mounted on glass slides. Sections were then washed on the slides 3 × 10 min. in PBS at RT.

Retinas were then incubated in Blocking Solution (2% donkey serum in 0.3% Triton (Millipore)) in PBS for 2 hr at RT. Retinas were then placed in Primary Antibody solution (Blocking Solution + 1:1000 rabbit anti-TH). Retinas were then washed 3 × 10 min. in PBS and placed in Secondary Antibody solution (Blocking Solution + 1:500 donkey anti-rabbit IgG 488). Retinas were then washed 3 × 10 min. PBS, covered in Fluoromount, coverslipped, and stored at 4°C until imaging.

## Imaging

For whole retina imaging, images were obtaining using Leica DM5500 SPE microscope at 100x under epifluorescent illumination. Tile scanned images were taken of the entire retina and automatically stitched together, using LAS X software.

Confocal images of retinal sections were obtained at 20X magnification using a Leica DM5500 SPE microscope.

## Cell counting

Retinal images were processed using ImageJ plugin: Fiji. Total TH positive cells in each retina were counted by hand by an experimenter blinded to the genotype, and total number was recorded.

## Dopamine quantification

Mice were euthanized at P14 and P60-90. Retinas were dissected and flash frozen using liquid nitrogen. Retinas were then placed on dry ice, stored at −80°C, and sent to the Vanderbilt University Neurochemistry Core on dry ice for analysis of dopamine levels. Samples where dopamine levels were compared directly were always quantified in the same batch to avoid inter-experiment variability.

## Statistics

Statistics were performed using Graphpad Prism software. For pairwise comparisons, unpaired t-tests were used. For multiple comparisons, a one-way ANOVA followed by a Dunnett's multiple comparisons test was performed. Significance was concluded when $p < 0.05$. n values represent number of retinas. Individual data points are shown on plots with a bar representing the mean.

## Acknowledgements

We would like to thank Jennifer Y Li for technical assistance with mouse husbandry and genotyping. We would like to thank Samer Hattar for the gift of the mouse lines used in this study. This work was funded by the Karl Kirchgessner Foundation Vision Research Grant and a Klingenstein-Simons Fellowship in the Neurosciences to TMS.

## Additional information

### Funding

| Funder | Author |
| --- | --- |
| Karl Kirchgessner Foundation | Tiffany M Schmidt |
| Esther A. and Joseph Klingenstein Fund | Tiffany M Schmidt |

The funders had no role in study design, data collection and interpretation, or the decision to submit the work for publication.

### Author contributions

Teona Munteanu, Conceptualization, Data curation, Formal analysis, Investigation, Writing—original draft, Writing—review and editing, Performed experiements, Analyzed data; Katelyn J Noronha, Data curation, Formal analysis, Writing—review and editing, Performed experiments; Amanda C Leung, Data curation, Investigation, Visualization, Writing—review and editing, Performed experiments; Simon Pan, Conceptualization, Data curation, Methodology, Writing—review and editing,

Performed critical plot experiments; Jasmine A Lucas, Conceptualization, Data curation, Writing—review and editing, Performed experiments; Tiffany M Schmidt, Conceptualization, Data curation, Formal analysis, Supervision, Funding acquisition, Investigation, Visualization, Methodology, Writing—original draft, Project administration, Writing—review and editing, Performed experiments, Analyzed data

### Author ORCIDs
Tiffany M Schmidt ⓘ https://orcid.org/0000-0002-4791-6775

### Ethics
Animal experimentation: This study was performed in accordance with the guidelines of the Animal Care and Use Committee at Northwestern University. (Protocol number IS00000887).

### Decision letter and Author response
Decision letter https://doi.org/10.7554/eLife.39866.008
Author response https://doi.org/10.7554/eLife.39866.009

## Additional files

### Supplementary files
• Transparent reporting form
DOI: https://doi.org/10.7554/eLife.39866.006

### Data availability
All data generated or analyzed during this study are included in the manuscript and supporting files. Individual data points are shown on each graph.

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
