## [Decision Letter]

Thank you for submitting your work entitled "Light-dependent pathways for dopaminergic amacrine cell development and function" for consideration by *eLife*. Your article has been reviewed by three expert reviewers, and the evaluation has been overseen by a Reviewing Editor and a Senior Editor. The reviewers have opted to remain anonymous.

The reviewers found the work interesting, but they have also raised a number of substantive questions. Based on the reviews and the follow-up discussion among the reviewers and the reviewing editor, we are requesting that you submit a revised manuscript. In particular, we think that the initial emphasis on ipRGCs is somewhat misplaced, given the outcome of the experiments.

A succinct summary of our assessment is as follows (this is a lightly edited comment from that discussion). "The scientific content reveals, simply, that the number of TH+ cells as well as DA content are both reduced following dark-rearing, and these effects are mediated by the rod pathway and independent of the ipRGCs. There is no clarification of the relationship between these two measures, and so the issue of DA function, so central to the Introduction, goes entirely unresolved. The issue of altered TH levels, rather than DA cell number is also a concern that the authors themselves raise. The assessment of DA cell morphology is not convincing. Overall, the manuscript is strong on making one point: that light, via the rod pathway, is critical for establishing the number of TH+ cells during development."

*Reviewer #1:*

Dopamine (DA) release in the retina is modulated by light exposure, affecting retinal circuitry that mediates our visual abilities under different light-adapted states. Recent studies have drawn attention to the role of intrinsically photosensitive RGCs (ipRGCs) as a possible mediator of the effects of light upon the DA cells, because of their anatomical and physiological relationship. The present manuscript sets the stage for addressing this issue, but in fact tackles another, that of whether light exposure during development affects the size of the DA cell population, and whether this effect is mediated by the ipRGCs, or instead by the rod or cone photoreceptors.

The authors find that dark rearing yields a significant reduction in the population of DA cells. The effect is detected as early as 2 weeks of age, shortly after eye opening, and is comparable to dark-rearing until maturity (P50-90). Returning mice to a standard L/D cycle at P14 until maturity did not return the number of DA cells to normal levels. A similar effect was achieved by comparing genetically engineered mice that lacked all ipRGC, rod and cone function (TKO); these too showed a reduction in the number of DA cells. In each comparison, the authors also assayed retinal DA content, and found a significant reduction as well.

To dissect which photosensitive cell is critical for mediating this light-dependency, the authors examined single KO mice lacking either ipRGC function, rod function or cone function, and repeated the analyses. DA cells and DA content were significantly reduced in only the mice lacking normal rod function; no significant effects were achieved in mice lacking phototransduction in either the ipRGCs or the cones. The authors confirmed that this rod function effect is not mediated through the ipRGCs, as ablating the latter during the first 2 postnatal weeks did not affect DA cell number nor content. Finally, the authors suggest that even in the absence of normal ipRGC number in this DTA mouse, the DA cells develop normally, evidenced by their characteristic stratification.

Major comments:

The inclusion of the analysis of DA content would, on the face of it, entitle the authors to make claims about light via rod-signaling regulating DA function, yet it is not clear whether the reductions in DA content are simply the knock-on consequence of fewer DA cells in each retina. So, one question that goes unanswered is whether the rod pathway is in fact a modulator of retinal dopamine levels, as concluded in the final sentence of the abstract. It seems surprising that the authors have not discussed this, but a closer examination of the data presented might suggest why: while the magnitude of the reduction in DA levels in Figure 1C is quite large relative to the magnitude of the effect upon DA cell numbers in Figure 1B caused by dark-rearing (suggesting that the former is not simply due to the reduction in cell number), a comparison between the TKO versus controls suggests a closer relationship. This is also suggested by the data comparing the RKO versus controls for these two measures. So, the authors may have concluded they cannot say anything definitive about rod-signaling during development ultimately affecting DA function in maturity independent of the reduction in DA cell numbers. The Introduction, in its concluding paragraph says, correctly, that they have tested whether light signaling via ipRGCs plays a role in DA cell development, yet then sums up by proclaiming it is the rod signaling that is responsible for the effect of light on DA cell development "and function". This latter stretch as indicated above, would seem to be for the purpose of bringing the results full circle, back to the fundamental issue of how light affects DA cell function (as opposed to how light exposure during development affects DA cell number and DA content).

The authors visually magnify the effects upon cell number by using a truncated Y axis beginning at 300 cells. It would be preferable for the authors to at least quote the means and sem within the text for each group, as well as speak to the magnitude of their effects (i.e. the size of the reduction, for both cell number and DA content). This is not to diminish in any way their solid claims showing an effect upon cell number-only that it is a very modest one. It also forces the consideration of whether the effect upon DA content might be solely explained by the reduction in DA cell number.

The authors point out the variability that can arise between different HPLC runs, ensuring that the controls for each comparison were run at the same time. This though cannot account for the variability in the magnitude of the effect between the different comparisons. By contrast, the number of DA cells is itself a more stable trait, though one that is known to vary across mice of different genetic backgrounds. This presumably accounts for the lower number of DA cells in the control condition in the DTA study, and the higher number in the control condition in the dark-rearing study, relative to the two replications of the control condition when comparing the TKO or the SKO to controls. Presumably all of the control mice used in these experiments are on distinct genetic backgrounds.

With respect to the argument that DA cells develop normally in the absence of ipRGCs, the degree of assessment is rather limited, confined to a qualitative examination of stratification patterns, though not of detailed single cell morphology or of the extent of the density of processes. If anything, the imaged cells from the DTA retinas would appear to show less processes outside of S1 of the IPL, implying stratification may very well be affected.

*Reviewer #2:*

Manteanu et al., have written a clear manuscript about the effects of rod-, cone-, and melanopsin-mediated pathways on the development of the number of dopaminergic neurons in mouse retina. Previously, the leading hypothesis was that melanopsin ganglion cells influenced the development of dopaminergic neurons because of their close functional association in mature retina. The authors use four transgenic lines that specifically knockout rod, cone, melanopsin, and all photoreceptor function. Quantification includes a count of tyrosine hydroxylase positive neurons in adult retina. The conclusion is that elimination of rod function decreases the total number of tyrosine hydroxylase-positive neurons; however, elimination of cone or melanopsin function alone is insufficient to change the number of TH-positive neurons.

General Comments:

A potentially major flaw with the interpretation of this manuscript is a point that the authors raise themselves in the last sentence of the Discussion section. The decrease in tyrosine hydroxylase-positive cells may not be equivalent to a decrease in the number of dopaminergic neurons if the TH levels have just decreased. A more general cell count number would help support the main finding of the manuscript. Are there alternative ways of labeling dopaminergic amacrine cells?

The rod transducin knockout eventually causes rod degeneration between 4-13 weeks (Calvert et al., 2000), which is the window over which cell counts are made. Have the authors considered the possibility that the rod manipulation causes photoreceptor degeneration and secondary cell death as a broad affect rather than a specific effect of the loss of rod function? The other cone and melanopsin knockouts are not known to cause retinal degeneration. This difference in retinal degeneration could potentially explain their results.

*Reviewer #3:*

The aim of the study by Munteanou et al., "Light-dependent pathways for dopamine amacrine cell (DAC) development and function," is to determine whether melanopsin-expressing intrinsically photosensitive ganglion cells (ipRGCs), which are known to control DAC activity, play a role in DAC development. The first set of experiments reveal that mice reared in constant darkness have a lower number of DACs and lower dopamine levels in the retina compared to mice reared on a normal light/dark cycle. Mice that lack functional photoreceptors (rods and cones and ipRGCs) have lower numbers of DACs and dopamine levels, similar to those in animals reared in constant darkness. Genetic elimination of cone function or ablation of ipRGCs has little effect on the elevated DAC number and dopamine content observed under normal rhythmic light/dark conditions. However, in mice that lack functional rods, the number of DACs and dopamine levels are low and similar to those measured in animals reared in constant darkness. The results suggest that normal light/dark cycles are required for the normal development of DACs and normal dopamine production. Surprisingly, the results further suggest that melanopsin and ipRGCs are not involved in the process and instead, rods (and cones?) seem to be required and sufficient.

Overall these findings provide evidence that conventional photoreceptors play a role in the control of DAC development and that surprisingly ipRGCs are not involved in this process. The results should be of broad interest to the scientific community including circadian biologists, vision scientists, as well as neuroscientists in general. The experiments are carefully done and interpreted appropriately (but see my comment on *Gnat2^cpfl3/cpfl3^*). The manuscript is well-written and the logic easy to follow. Overall, this is a well-executed study that reports exciting new findings with significant ramifications.

1) Throughout the manuscript, the notion that "light exposure is required to set the number of TH+ cells and dopamine levels in the retina" is vague and somewhat misleading. What is important here is the role of prolonged exposure to light every day for 12 hours. That is, a normal light/dark cycle is required for normal DAC development, as opposed to constant darkness. This should be clarified.

2) Results section: The authors should give more information about the timeline of the experiments in which animals were placed in DD and subsequently in LD and in particular, when these animals were euthanized. Also, in the Figure 1 legend, please define adult age. This information is the Materials and methods section but should be stated in the main text as well.

3) Results section: Correct the typo in *Gnat2^cpfl3/cpfl(2)3^*. The authors should be cautious with their interpretation of the *Gnat2^-/-^* results: *Gnat2^cpfl3/cpfl3^*mice do have cone function but with a much lower sensitivity. The light intensity used in this study (app. 2,000 lux) is well above cone threshold. Therefore, the authors cannot exclude the contribution of cones in DAC development. Still, the main conclusion of the paper remains unchanged: ipRGCs and melanopsin are not involved in the control of DAC development by light.

---

## [Author Response]

A succinct summary of our assessment is as follows (this is a lightly edited comment from that discussion). "The scientific content reveals, simply, that the number of TH+ cells as well as DA content are both reduced following dark-rearing, and these effects are mediated by the rod pathway and independent of the ipRGCs. There is no clarification of the relationship between these two measures, and so the issue of DA function, so central to the Introduction, goes entirely unresolved. The issue of altered TH levels, rather than DA cell number is also a concern that the authors themselves raise. The assessment of DA cell morphology is not convincing. Overall, the manuscript is strong on making one point: that light, via the rod pathway, is critical for establishing the number of TH+ cells during development."

As mentioned in the initial paragraph, we distilled this summary into three main points based on our understanding of the paragraph above. Our responses and accompanying changes are detailed below.

1) Change the focus of the Introduction to better reflect the findings reported in the manuscript.

The reviewers make an excellent point that the Introduction emphasizes ipRGC function, but that the data clearly indicate an important role for rods via an ipRGC-independent pathway. We have edited the Introduction to better introduce how each of the three photoreceptors provides input to DACs. We do think it is important to emphasize the fact that ipRGC-dependent influences was a prevailing hypothesis prior to this work, but hope that the Introduction now provides more complete and relevant background information. We thank the reviewers for this suggestion as it enhances the clarity of the manuscript and better prepares the reader for the upcoming data.

*2) It is unclear whether reduced* TH+ *cell number is a function of cell loss or decrease in levels of TH within the cells.*

One limitation of this study is our inability to determine whether DACs are actually lost or whether TH levels are decreased leading to lack of detection of DACs. This is an issue that we are keen to overcome, but we have not able to identify reagents that would let us address this issue with interpretable results. For example, mouse lines that would permanently label DACs (DAT-Cre, TH-Cre) label multiple, additional (and not entirely overlapping) populations in the retina(1). Additionally, the TH-RFP reporter line labels additional retinal cell types beyond the canonical DACs and is dependent on TH levels for reporter expression. Similarly, any cell death analysis would continue to rely on TH expression as a proxy for the presence of DACs, and is therefore fraught with the same limitations or interpretation of the results as immunolabeling for TH. We have therefore expanded our discussion of this issue within the manuscript.

3) Better illustrate DAC morphology in Opn4DTA versus WT retinas.

The reviewers have pointed out that, as presented, the morphology of WT and Opn4^DTA^TH+ cells might be construed to be different. We thank the reviewers for bringing this to our attention. We have included 10 additional examples of each genotype as a supplemental figure (Figure 2—figure supplement 1) to demonstrate the natural variability of the TH+ process stratification and more clearly demonstrate that our examples fall in the normal range. This is referenced as Figure 2—figure supplement 1 in the text.

Reviewer #1:Dopamine (DA) release in the retina is modulated by light exposure, affecting retinal circuitry that mediates our visual abilities under different light-adapted states. Recent studies have drawn attention to the role of intrinsically photosensitive RGCs (ipRGCs) as a possible mediator of the effects of light upon the DA cells, because of their anatomical and physiological relationship. The present manuscript sets the stage for addressing this issue, but in fact tackles another, that of whether light exposure during development affects the size of the DA cell population, and whether this effect is mediated by the ipRGCs, or instead by the rod or cone photoreceptors.

We thank the reviewer for pointing out the incongruity between the content of our intro and the results obtained in the study. We have edited the Introduction to better reflect the upcoming results (see Point 1 above).

[…] Major comments:The inclusion of the analysis of DA content would, on the face of it, entitle the authors to make claims about light via rod-signaling regulating DA function, yet it is not clear whether the reductions in DA content are simply the knock-on consequence of fewer DA cells in each retina. So, one question that goes unanswered is whether the rod pathway is in fact a modulator of retinal dopamine levels, as concluded in the final sentence of the abstract. It seems surprising that the authors have not discussed this, but a closer examination of the data presented might suggest why: while the magnitude of the reduction in DA levels in Figure 1C is quite large relative to the magnitude of the effect upon DA cell numbers in Figure 1B caused by dark-rearing (suggesting that the former is not simply due to the reduction in cell number), a comparison between the TKO versus controls suggests a closer relationship. This is also suggested by the data comparing the RKO versus controls for these two measures. So, the authors may have concluded they cannot say anything definitive about rod-signaling during development ultimately affecting DA function in maturity independent of the reduction in DA cell numbers. The Introduction, in its concluding paragraph says, correctly, that they have tested whether light signaling via ipRGCs plays a role in DA cell development, yet then sums up by proclaiming it is the rod signaling that is responsible for the effect of light on DA cell development "and function". This latter stretch as indicated above, would seem to be for the purpose of bringing the results full circle, back to the fundamental issue of how light affects DA cell function (as opposed to how light exposure during development affects DA cell number and DA content).

We removed the phrase “and function” from the last line of the Introduction and from the title of Figure 3.

Regarding the differences in DD/LD versus the genetic manipulations: It is important to note that the light intensities used in the LD versus DD condition were brighter (~2000 lux where the animals were all reared in light tight boxes) than those used for the mutant lines (~100-200 lux, where the animals were reared in the animal facility). It is therefore possible that the increased light intensity results in higher dopamine levels and so the proportional decrease between LD (2000 lux) versus DD is more dramatic than for the other conditions. The DA measurements were run in different batches, and so the absolute DA levels cannot be directly compared between the two experiments. We have also expanded our discussion of the issue of whether the decreased DA levels are solely the result of lower TH+ cell number or whether there is also potentially a decrease in TH levels in the remaining TH+ cells.

The authors visually magnify the effects upon cell number by using a truncated Y axis beginning at 300 cells. It would be preferable for the authors to at least quote the means and sem within the text for each group, as well as speak to the magnitude of their effects (i.e. the size of the reduction, for both cell number and DA content). This is not to diminish in any way their solid claims showing an effect upon cell number-only that it is a very modest one. It also forces the consideration of whether the effect upon DA content might be solely explained by the reduction in DA cell number.

We thank the reviewer for this suggestion and have added the numbers to the text as requested. We have also expanded our discussion on the point of whether the DA decrease is due solely to loss of DACs themselves or potentially also due to decreased DA production by the remaining TH+ cells.

The authors point out the variability that can arise between different HPLC runs, ensuring that the controls for each comparison were run at the same time. This though cannot account for the variability in the magnitude of the effect between the different comparisons. By contrast, the number of DA cells is itself a more stable trait, though one that is known to vary across mice of different genetic backgrounds. This presumably accounts for the lower number of DA cells in the control condition in the DTA study, and the higher number in the control condition in the dark-rearing study, relative to the two replications of the control condition when comparing the TKO or the SKO to controls. Presumably all of the control mice used in these experiments are on distinct genetic backgrounds.

We did indeed use littermate controls for each experiment in order to prevent any differences due to variability of genetic background. Each of our mouse lines is on a hybrid C57Bl6/129 hybrid background, meaning that the controls of different strains could indeed have slightly different genetic backgrounds, making the littermate control condition doubly important.

With respect to the argument that DA cells develop normally in the absence of ipRGCs, the degree of assessment is rather limited, confined to a qualitative examination of stratification patterns, though not of detailed single cell morphology or of the extent of the density of processes. If anything, the imaged cells from the DTA retinas would appear to show less processes outside of S1 of the IPL, implying stratification may very well be affected.

We thank the reviewer for pointing out that our images suggest differences between the WT and the DTA lines. We now include 10 additional WT and DTA examples that demonstrate the natural variability in the stratification of TH+ processes in both of these lines. We include these data as a supplemental figure (Figure 2—figure supplement 1) to illustrate better similarity to the WT image in the main figure and have swapped out one of the DTA examples in the main Figure 2.

Reviewer #2:[…] General comments:A potentially major flaw with the interpretation of this manuscript is a point that the authors raise themselves in the last sentence of the Discussion section. The decrease in tyrosine hydroxylase-positive cells may not be equivalent to a decrease in the number of dopaminergic neurons if the TH levels have just decreased. A more general cell count number would help support the main finding of the manuscript. Are there alternative ways of labeling dopaminergic amacrine cells?

We agree with the reviewer that we are unable to resolve whether the cells have been completely lost or are simply not producing TH at detectable levels. For the reasons mentioned in response to the general summary, we are unable to identify a way to label this population of cells reliably as available reagents either label additional amacrine cells in the retina or continue to rely on TH expression. However, it is important to note that the decrease in TH does indicate that even if the cells themselves are still present, they are not expressing an enzyme critical for DA synthesis and therefore, they are effectively no longer “dopaminergic.” Thus, though we cannot clarify the reasons for the absence of TH immunolabeling, it is clear that dopamine production from a subset of DACs is lost or at least severely attenuated in dark reared mice or mice lacking rod phototransduction.

The rod transducin knockout eventually causes rod degeneration between 4-13 weeks (Calvert et al., 2000), which is the window over which cell counts are made. Have the authors considered the possibility that the rod manipulation causes photoreceptor degeneration and secondary cell death as a broad affect rather than a specific effect of the loss of rod function? The other cone and melanopsin knockouts are not known to cause retinal degeneration. This difference in retinal degeneration could potentially explain their results.

This is an important point and we thank the reviewer for bringing it to our attention. We don’t think that this explains the effect because we are able to recapitulate the phenotype in dark-reared animals where there is not retinal degeneration (2). Nonetheless, we have added a discussion of these two important points to the Discussion section.

Reviewer #3:

*[…] 1) Throughout the manuscript, the notion that "light exposure is required to set the number of* TH+ *cells and dopamine levels in the retina" is vague and somewhat misleading. What is important here is the role of prolonged exposure to light every day for 12hours. That is, a normal light/dark cycle is required for normal DAC development, as opposed to constant darkness. This should be clarified.*

We thank the reviewer for pointing out the vague language and have attempted to clarify the language at the beginning of the results to say “we first tested whether exposure to a 12:12 light: dark cycle influences the number of TH+ amacrine cells in the retina” to highlight this aspect of the experimental design.

2) Results section: The authors should give more information about the timeline of the experiments in which animals were placed in DD and subsequently in LD and in particular, when these animals were euthanized. Also, in the Figure 1 legend, please define adult age. This information is the Materials and methods section but should be stated in the main text as well.

This is an excellent point. We have added altered the description in the opening paragraph of the results to read, “[…] in the retinas of animals that were reared in constant darkness from conception” to clarify that animals bred, and their pups reared in constant darkness. We have also added the term “from conception” to the description in Figure 1.

3) Results section: Correct the typo in Gnat2^cpfl3/cpfl(2)3^. The authors should be cautious with their interpretation of the Gnat2^-/-^ results: Gnat2^cpfl3/cpfl3^ mice do have cone function but with a much lower sensitivity. The light intensity used in this study (app. 2,000 lux) is well above cone threshold. Therefore, the authors cannot exclude the contribution of cones in DAC development. Still, the main conclusion of the paper remains unchanged: ipRGCs and melanopsin are not involved in the control of DAC development by light.

We thank the reviewer for bringing this important point to our attention. We have added a discussion of this point to the Discussion section.